# OpenFilter: A Framework to Democratize Research Access to Social Media AR Filters

**Piera Riccio**[1]  **Bill Psomas**[2,6]  **Francesco Galati**[3]  **Francisco Escolano**[4]

**Thomas Hofmann**[5]  **Nuria Oliver**[1]

[1]ELLIS Alicante  [4]Universidad de Alicante  [3]EURECOM  [5]ETH Zurich

[2]Institute of Advanced Research in Artificial Intelligence (IARAI)

[6]National Technical University of Athens

`piera@ellisalicante.org`

## Abstract

Augmented Reality or AR filters on selfies have become very popular on social media platforms for a variety of applications, including marketing, entertainment and aesthetics. Given the wide adoption of AR face filters and the importance of faces in our social structures and relations, there is increased interest by the scientific community to analyze the impact of such filters from a psychological, artistic and sociological perspective. However, there are few quantitative analyses in this area mainly due to a lack of publicly available datasets of facial images with applied AR filters. The proprietary, close nature of most social media platforms does not allow users, scientists and practitioners to access the code and the details of the available AR face filters. Scraping faces from these platforms to collect data is ethically unacceptable and should, therefore, be avoided in research. In this paper, we present OPENFILTER, a flexible framework to apply AR filters available in social media platforms on existing large collections of human faces. Moreover, we share FAIRBEAUTY and B-LFW, two beautified versions of the publicly available FAIRFACE and LFW datasets and we outline insights derived from the analysis of these beautified datasets.

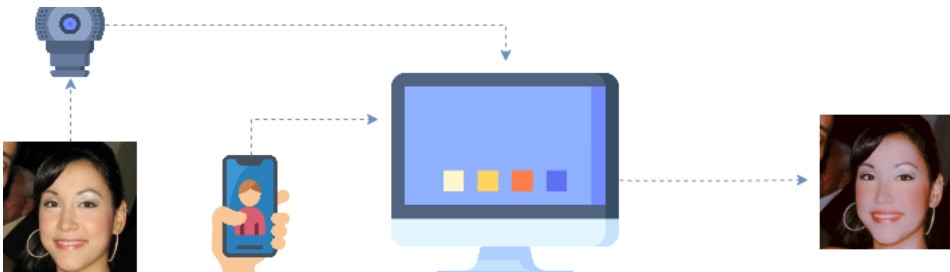

Figure 1: OPENFILTER pipeline. A machine runs the targeted social media application (e.g. Instagram) on an Android emulator. An image from the dataset is projected on the camera opened through the social media application. A filter is directly applied to the image. This Figure has been designed using resources from Flaticon[1] and [30].

---

[1]https://www.flaticon.com/

36th Conference on Neural Information Processing Systems (NeurIPS 2022) Track on Datasets and Benchmarks.

# 1 Introduction

Selfies (photos of oneself, typically captured with smartphones or webcams) have become very popular on social media platforms, particularly on Instagram[2], Snapchat[3] and TikTok.[4] Google reported that its Android devices took 93 million selfies per day in 2019 and in 2021 Instagram users uploaded an average of 95 million photos and 250 million stories each day [9]. In a recent poll, 18-to-24-year-old participants reported that every third photo they take is a selfie [56]. In fact, selfies constitute a new visual genre [10] that centralizes self-expression in its most traditional interpretation: presenting a positive view of the self, conforming to social norms and meeting the expectations of others to receive positive feedback [20]. AI-enhanced face filters are becoming increasingly pervasive on social media platforms [17]. These filters leverage algorithmic advances in Computer Vision to automatically detect the face and facial features of the user, and Computer Graphics (namely Augmented Reality or AR) to superimpose digital content in real-time, enhancing or distorting the original facial image [44]. Hence, we shall refer to them as *AR filters*. In their original form, selfies were conceived as a digital manifestation of an underlying reality (i.e. the face of a person), and its relation to the place and space [26]. However, considering the wide adoption of these filters, the original equation relating selfies to real human faces adopts new dynamics and the online identity becomes an artifact.

Different types of AR face filters are currently available on social media for a variety of applications and use cases, including marketing and commercials [3], entertainment, and aesthetics [18]. Users of social media platforms are able to create and share their own AR filters using off-the-shelf tools, establishing a new creative form of expression and a new artistic role: the *filter creator*. Other users can use the filters to experience different versions of themselves, with e.g. futuristic or sci-fi scenarios, funny deformations, horrifying or surreal textures, 3D makeup or beauty enhancement. The possibility of experiencing these transformations transcends the physical location of the users as all that is needed is a smartphone with an Internet connection, making AR face filters a form of post-Internet art [4]. The COVID-19 pandemic has represented a turning point for the general acceptance of AR filters as an effective form of art [25], giving filter creators an essential responsibility in shaping the cultural and societal impact of this technology.

Given the importance of faces in our social structures and relations, and the wide adoption of AR face filters, the scientific community has shown increased interest to analyze the impact of such filters from a psychological, artistic and sociological perspective [37]. However, there are few quantitative analyses in this area mainly due to a lack of publicly available datasets of facial images with applied AR filters. The proprietary, close nature of most social media platforms does not allow users, scientists and practitioners to access the code and the details of the available AR face filters [24]. Scraping faces from these platforms to collect data is ethically unacceptable and should, therefore, be avoided in research. A possible solution to this challenge consists of recruiting volunteers to participate in user studies to create a dataset with their content after obtaining their informed consent. However, this approach is time-consuming, expensive and non-scalable. In this paper, we provide a methodology to overcome these limitations and democratize access to AR filters used in social media for research purposes.

Specifically, we make the following contributions:

1. We present OPENFILTER (section 3), a flexible open framework to apply AR filters available in social media platforms on existing, publicly available large collections of human faces.

2. In our case study (section 4), we focus on filters intended to enhance facial aesthetics. In this regard, we share FAIRBEAUTY and B-LFW, the beautified versions of the publicly available FAIRFACE [30] and LFW [27] datasets.

3. We conduct face similarity and recognition experiments on these beautified datasets and outline several insights from a technical and sociological perspective (section 5).

---

[2]https://www.instagram.com/
[3]https://www.snapchat.com/
[4]https://www.tiktok.com/

## 2    Related Work

The popularity of AR filters on social media has led to an increased interest in the research community towards understanding their impact from a variety of perspectives. However, there is a lack of publicly available datasets to enable a quantitative, systematic analysis of filtered face images [24]. Hence, user studies and surveys are the most commonly explored techniques to quantify the impact of these filters, overcoming the legal and ethical implications related to the direct scraping of the data from social media. Early work by [17] studied the impact of face filters on self-perception and self-esteem through a user study including 33 participants (23 females and 10 males). In this work, the authors highlight that the self-perception of the body image is highly influenced by social factors and that we tend to assume that attractive features in others are also desirable in ourselves, especially in individuals with low self-esteem.

More recently, [18] studied the short-term perception of users on their appeal, personality, intelligence and emotion when different distortions are applied through AR face filters. The study included 18 different SnapChat filters on 20 male and 20 female users, with ages ranging from 20 to 50 years old, and mostly white people. The authors report that people self-perceive the targeted characteristics in their facial traits in the same way as they perceive them in others. They conclude that even small changes in the facial traits impact self-recognition capabilities and that the eyes are particularly relevant for conveying emotions.

Surveys and user studies are undoubtedly valuable methodologies to address certain research questions. However, they are hardly scalable and hence do not enable carrying out quantitative studies of the impact of these filters. To mitigate this limitation, scholars have approximated the AR filters available on social media through alternative techniques or software. In particular, [6] introduce a database of beautified faces with three different ethnic variations. The dataset is generated using Fotor[5], BeautyPlus[6] and PortraitPro Studio Max[7] and it is composed of 600 different individuals of three different ethnicities (Indian, Chinese and Caucasian) with balanced genders. Note that the word *beautified* refers to a set of digital retouching techniques, including skin smoothing, skin tone enhancement, acne removal, face slimming, eye and lip color change, and distortion of jaw and forehead. More recently, [24] create a beautified version of LFW [27] benchmark dataset. In their case, the word *beautified* refers to the superimposition of simple AR elements (e.g. dog nose, transparent glasses, sunglasses) on the original faces. While not focused on beautification, [35] study the effect of surgical masks on face recognition techniques, creating mask overlays using SparkAR.[8]

These methods are able to apply simple filters (such as the superimposition of glasses, or masks) on pre-existing images. However, they are unable to reproduce the intrinsic cultural and sociological value of the user-generated filters available on social media, including the beauty filters. In this work, we address the limitations of the aforementioned methods by providing a framework called OPENFILTER to apply *any* AR filter directly obtained from social media applications to pre-existing face datasets. OPENFILTER enables the creation of large-scale datasets that represent the current cultural ecosystems on social media platforms. Note that an approximation of these filters through other methodologies risks biasing the study and jeopardizing the ecological validity of the results.

## 3    OPENFILTER: A Framework for AR-based Filtered Dataset Creation

Most of the AR filters available on social media platforms –such as Instagram, TikTok, SnapChat– can only be applied in real-time on selfie images captured from the camera of the smartphone. Hence, it is challenging to carry out quantitative and systematic research on such filters. OPENFILTER fulfills such a need by enabling the application of AR filters on publicly available datasets of faces. The pipeline architecture of OPENFILTER is depicted in Figure 1 and the code is available in our repository[9].

OPENFILTER allows the application of AR filters directly from social media through (1) an Android Emulator, (2) a machine and (3) a virtual webcam. The Android emulator runs on the machine, where

---

[5] https://www.fotor.com/es/
[6] https://play.google.com/store/apps/details?id=com.commsource.beautyplus
[7] https://www.anthropics.com/
[8] https://sparkar.facebook.com/ar-studio/
[9] https://github.com/ellisalicante/OpenFilter

the social media application targeted in the research is installed[10]. In the emulator, the researcher may access any available AR filter of the social media platform. As previously stated, most of these filters can only be applied to live images from the camera. To overcome this limitation, the virtual webcam projects the existing image dataset on the camera enabling the application of the AR filters on it. Through an auto-clicker system, each image is first projected on the camera; next, the filter is applied to the image and finally the filtered image is saved on disk. The instructions for use and a walk-through video are available in our repository; an exemplary screenshot and code snippets can be found in the Appendix. OPENFILTER processes an image every 4 seconds on a Intel(R) Core(TM) i7-8565U machine with NVIDIA GeForce MX150, i.e. around 900 images per hour and 22,000 per day.

Next, we describe two novel datasets created using OPENFILTER by applying eight popular AR beautification filters to the FAIRFACE [30] and LFW [27] benchmark face datasets. We also provide insights derived from the analysis of the impact of the beauty filters on the original face images.

## 4    FAIRBEAUTY and B-LFW: Two Novel Datasets of Beautified Faces

In recent years, AR face filters are increasingly used to beautify the original faces and make them conform to certain canons of beauty by digitally modifying facial features, especially among female users [45]. We shall refer to these filters as *beauty filters*. While selfies have been used to challenge beauty norms and to propose different and ironic perspectives [15, 2, 55], the popularity of beauty filters seems to be pushing in the opposite direction. According to [14], beauty filters contribute to the sexualization of women, while [16] claim that the aesthetic concept behind beauty filters projects the female faces closer to normative ideals of femininity.

Nowadays, thousands of AR beauty filters are available on Instagram and other social media platforms. These filters apply similar transformations to the input image: they provide a smooth and uniformly colored skin, almond shaped eyes and brows, full lips, a small nose and a prominent cheek structure [46, 38, 33, 29, 50, 22, 41] (see the examples in Figure 2). Given the scale of this phenomenon, beauty filters are an interesting research topic to strengthen our understanding of the development of contemporary culture and aesthetics [49]. On the one hand, the extensive use and exposure to beautified selfies seem to be a homogenizing force of beauty standards, contributing to a significant increase in teenage plastic surgery [31] and mental health issues [1]. On the other, today's fashion brands, companies, magazine editors and movie producers are encouraging a more diverse and inclusive view of beauty, which is partly attributed to the emergence and persistence of the selfies culture [19]. In this paper, we do not take a stand on the virtues and dangers of beauty filters for society: as with every technology, its use creates a spectrum of possibilities whose value should be properly investigated and understood through interdisciplinary research. For this reason, we share a technical tool (OPENFILTER) that facilitates quantitative and qualitative research in this field and present an exemplary case study on two novel datasets: FAIRBEAUTY and B-LFW.

FAIRBEAUTY is a beautified version of the FAIRFACE dataset [30]. FAIRFACE (license CC BY 4.0) contains 108,501 face images, promoting algorithmic fairness in Computer Vision systems. The choice of this dataset is motivated by its focus on *diversity* and our will to identify a dataset that would be representative of the population of Instagram –which is a globalized social environment with over 800 million users in the world[11]– without biasing the results towards specific facial traits, gender or age ranges. In FAIRBEAUTY, eight popular, AR beauty filters are applied on equal portions of the original dataset. An example of the applied filters is shown in Figure 2. The choice of the beauty filters is based on their popularity, which we assessed through articles in women's magazines[12] and relevant trends on Instagram. All selected beauty filters have been created by Instagram users that describe themselves as filter/digital artists.

B-LFW is a beautified version of the LFW (Labeled Faces in the Wild) [27] dataset, a public benchmark dataset for *face verification*, designed for studying and evaluating unconstrained face recognition systems. This dataset contains more than 13,000 facial images of 1,680 different

---

[10]In our implementation, we refer to Instagram, but OPENFILTER may be used with any other social media application available on the Android emulator.

[11]https://www.statista.com/statistics/578364/countries-with-most-instagram-users/

[12]https://creatorkit.com/blog/most-popular-instagram-filters-effects and
https://inflact.com/blog/instagram-filters-for-stories/, accessed in April 2022

individuals who appeared in the news and hence are public figures. In this work, we have beautified LFW with the same eight popular Instagram beauty filters described above and depicted in Figure 2, using different filters on different images from the same individuals.

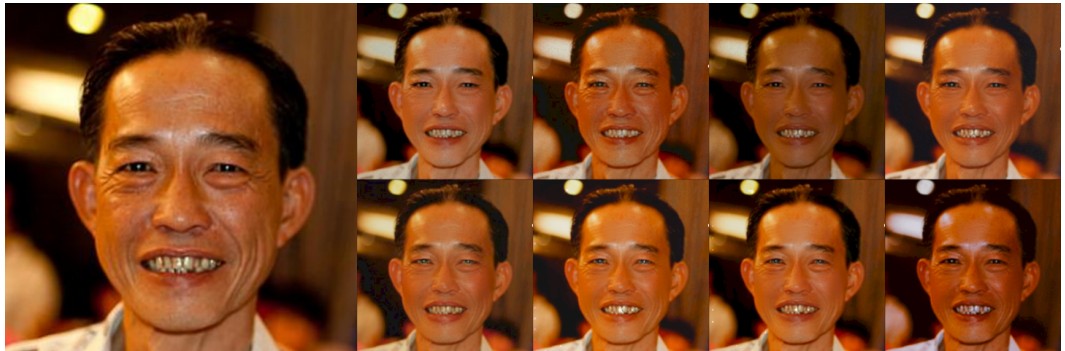

Figure 2: Example of the eight different beauty filters applied to the left-most image [30]. From left to right and top to bottom: filter 0 -*pretty* by *herusugiarta*; filter 1 -*hari beauty* by *hariani*; filter 2 -*Just Baby* by *blondinochkavika*; filter 3 -*Shiny Foxy*, filter 4 -*Caramel Macchiato* and filter 5 -*Cute baby face* by *sasha_soul_art*; filter 6 -*Baby_cute_face_* by *anya__ilicheva*; filter 7 -*big city life* by *triutra*.

## 4.1 Intended Use

In this paper, we address two research questions through the analysis of the shared datasets. In **RQ1**, we investigate whether beauty filters *homogenize faces*, hence reducing the distance between pairs of faces. In **RQ2**, we explore the impact of beautified faces on *face verification* techniques. In addition, we outline other research directions that would be enabled by the beautified face datasets.

The first direction concerns studying the influence of beauty filters on social constructs, such as trustworthiness, homophily and intelligence, both computationally and through user studies. Recent work has reported that humans trust deep fakes more than faces of real humans [39] and scholars have found an *attractiveness halo* [13, 53], which is the tendency to assign positive qualities and traits –such as higher morality, better mental health, and greater intelligence– to physically attractive people. The two datasets shared in this paper would enable empirical research to assess the existence, prevalence and intensity of the attractiveness halo.

A second direction concerns the societal implications of beauty filters on social media platforms. These filters have raised concerns regarding existing biases in the automatic beautification practices and have been widely criticized for perpetuating racism [38] and, in particular, colorism [46]. Note that the filters we apply to obtain the beautified datasets are selected due to their popularity on social media platforms, without considering the cultural background of the filter creators. FAIRBEAUTY opens the possibility of studying such issues computationally, and understanding their nature and scope.

We strongly *discourage* controversial and unethical uses of our framework and datasets, including the development of beautification removal applications. In 2017, a Make-Up Remover App[13] was released, unleashing a wave of criticism [32, 5, 34] as it was perceived as sexist and misogynistic. We acknowledge that the removal of beauty filters may be considered an insightful research topic from a technical perspective, and some of the application fields (e.g. psychotherapy for teenagers dealing with low self-esteem and dysmorphia) could be highly beneficial. However, the wide distribution of such a tool to the general public could have negative unintended effects. In addition, regarding the development of face recognition techniques, we stress that this technology raises several legal and ethical challenges [11], which need to be taken into account to avoid perpetuating injustice [43] and to preserve the privacy of individuals [11]. Considering these potential implications, we share all our assets with exclusively non-commercial licenses (CC BY-NC-SA 4.0 for the datasets, dual licensing of GNU General Public License version 2 for OpenFilter), encouraging our readers to be always cognizant of the implications of their uses.

---

[13]The application is no longer available.

In the next section, we describe two preliminary experiments on the generated datasets to address RQ1 and RQ2. We study RQ1 –the homogenization of faces– on the FAIRBEAUTY dataset. We explore RQ2 –the impact of beauty filters on face recognition systems– by analyzing the B-LFW dataset.

# 5 Experiments

## 5.1 Preliminaries

**Problem formulation** We are given an evaluation set $X \subset \mathcal{X}$, where $\mathcal{X}$ is the input space, and a transformation set $\mathcal{T}$. We are also given a model $f_\theta : X \to \mathbb{R}^d$ that maps input samples to a $d$-dimensional embedding vector. Parameters $\theta$ are obtained, as $f$ is typically pre-trained on a larger set. Given two sample images $x, x' \in \mathcal{X}$, we denote by $d(x, x')$ the distance between $x, x'$ in the embedding space, typically an increasing function of Euclidean distance. We call $x, x'$ a pair. The set $\mathcal{T}$ contains transformations shown in Figure 3, such as beautification, Gaussian filtering or down-sampling. We denote these transformations by $t_b, t_g, t_s \in \mathcal{T}$ respectively. We denote by $x_b$ the beautified version of $x$, that is $x_b = t_b(x)$, etc; $t_g^{\sigma=n}$ represents the application of a Gaussian filter with radius $n$ on image $x$, which will result in an image $x_g$, while $t_s^{w,h=N}$ represents the down-sampling from $\mathbb{R}^{H \times W \times 3} \to \mathbb{R}^{N \times N \times 3}$, which will result in an image $x_s$. It is common to $\ell_2$-normalize the embeddings. To simplify the notation, we drop the dependencies of $f, d$.

**Setup** We conduct experiments leveraging different face verification models to determine the similarity between pairs of faces. Three of them –namely DeepFace [52], VGG-Face [40], and Facenet [47]– are well-known models available in the Python library `deepface` [48]; the other three – CurricularFace [28], MagFace [36], and ElasticFace [8]– are recent state-of-the-art models for face recognition. DeepFace and VGG-Face use a custom CNN architecture with an embedding size $d = 4096$, Facenet uses Inception-ResNet [51] with an embedding size $d = 128$. CurricularFace, MagFace and ElasticFace use ResNet100 [12] with an embedding size $d = 512$. DeepFace, VGG-Face and Facenet are pre-trained on the VGGFace2 dataset [40], while CurricularFace, MagFace and ElasticFace are pre-trained on the MS1MV2 dataset [12], a refined version of the MS-Celeb-1M [21], containing 5.8M images of 85k identities. We evaluate on both original and transformed datasets following the evaluation protocols and metrics of each dataset.

## 5.2 Experiments on FAIRBEAUTY: RQ1 - Do beauty filters homogenize faces?

The AR beauty filters detect the position of the faces in an original image and super-impose digital content to modify (i.e. to *beautify*) the original facial features. As these filters apply the same transformation to the facial features of all faces, we hypothesize that they homogenize facial aesthetics making the beautified faces more similar to each other. As previously stated, the images in FAIRFACE are diverse by design. In this experiment, we aim to assess whether the application of beauty filters reduces the diversity, i.e. it homogenizes the FAIRFACE dataset.

To determine the homogenization of the filtered faces, we consider both the FAIRFACE and the FAIRBEAUTY datasets. We conduct this experiment using the six different models previously described, i.e. DeepFace, VGG-Face, Facenet, CurricularFace, MagFace and ElasticFace. First, we sample pairs of faces. Next, we forward them through a pre-trained model $f$ and obtain the corresponding embedding vectors to compute the distance $d$ between them. For every experiment, we compute the distances between a different subset of 500 pairs of images, so that the overall measurements consider 3,000 distinct pairs of images, to minimize potential biases in the results. We evaluate the homogenization using the average distance of all sampled pairs from FAIRFACE and FAIRBEAUTY datasets, i.e. the lower the average distance, the greater the homogenization. In FAIRBEAUTY, the eight selected beauty filters are applied on equal portions of the original FAIRFACE dataset, to better simulate a social media scenario. Note that the images are selected without considering the applied filter, and the loss of diversity is therefore analyzed even when applying different beauty filters to different images that are compared. As a reference, we perform the same computation when applying Gaussian filtering (blurring) and down-sampling (pixelation) to the original faces of the FAIRFACE dataset. This comparison allows a better understanding of the potential diversity loss due to the beauty filters. Examples of the original, beautified, Gaussian filtered and down-sampled images are shown in Figure 3, while the algorithm can be found in Algorithm 1.

---

**Algorithm 1** Computation of pair-wise face distances

---
**Require:** Datasets FAIRFACE, FAIRBEAUTY, Model $f$
**Ensure:** Collection $C$

1: $C \leftarrow \{\}$
2: **repeat**
3:     Sample $(\mathbf{x}, \mathbf{x}')$ from FAIRFACE
4:     Select $(\mathbf{x}_b, \mathbf{x}'_b)$ from FAIRBEAUTY
5:     $\mathbf{x}_g, \mathbf{x}'_g \leftarrow t_g^{\sigma=2}(\mathbf{x}), t_g^{\sigma=2}(\mathbf{x}')$
6:     $\hat{\mathbf{x}}_g, \hat{\mathbf{x}}'_g \leftarrow t_g^{\sigma=3}(\mathbf{x}), t_g^{\sigma=3}(\mathbf{x}')$
7:     $\mathbf{x}_s, \mathbf{x}'_s \leftarrow t_s^{w,h=64}(\mathbf{x}), t_s^{w,h=64}(\mathbf{x}')$

8:     $m \leftarrow ||f(\mathbf{x}) - f(\mathbf{x}')||_2$
9:     $\Delta_b \leftarrow ||f(\mathbf{x}_b) - f(\mathbf{x}'_b)||_2 - m$
10:    $\Delta'_g \leftarrow ||f(\mathbf{x}_g) - f(\mathbf{x}'_g)||_2 - m$
11:    $\Delta''_g \leftarrow ||f(\hat{\mathbf{x}}_g) - f(\hat{\mathbf{x}}'_g)||_2 - m$
12:    $\Delta_s \leftarrow ||f(\mathbf{x}_s) - f(\mathbf{x}'_s)||_2 - m$
13:    $C \leftarrow C \cup \{\Delta_b, \Delta'_g, \Delta''_g, \Delta_s\}$
14: **until** 500 repetitions are reached

---

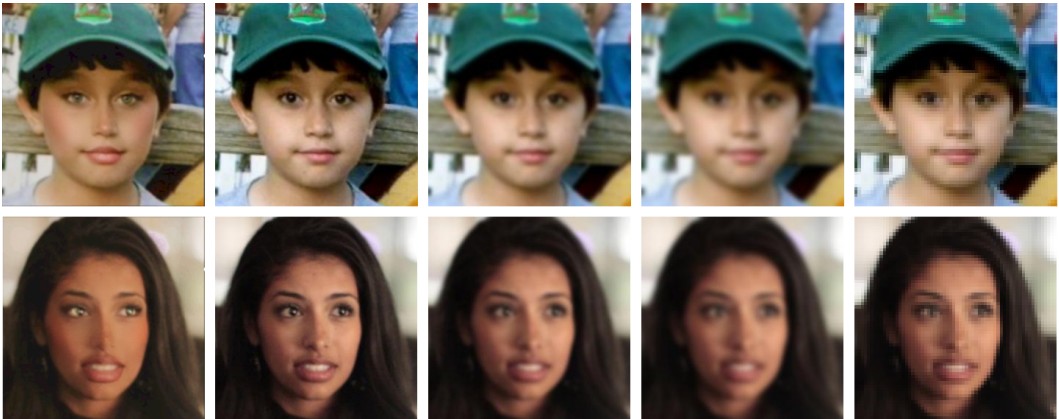

Figure 3: An exemplary pair of images from [30] illustrating the five different versions that are analyzed to address RQ1: the face homogenization experiment. From left to right: beautified version using OPENFILTER, original version, blurred version with Gaussian filter at radius 2, blurred version with Gaussian filter at radius 3, down-sampled (pixeled) version to 64x64 pixels.

The results of this experiment are shown in Figure 4. For each pair, distances between transformed images are plotted in terms of differences w.r.t. the distance between the original images. A value of 0 (plotted as a dashed red line in the Figure) means that there is no difference between the original distance and the distance after applying one transformation, i.e. the transformation does not affect the distance between the faces. In Figure 4, we observe a significant difference in the distances between the original and the transformed faces. Depending on the experiment, the reduction in distances that comes with beautification is comparable to the effect of applying either Gaussian filters or down-sampling on the images. In all cases, the measurements obtained on the beautified version have lower average distance than those of the original dataset. In other words, according to these experiments, the beautified faces in FAIRBEAUTY are statistically more similar to each other than the original faces. Thus, the answer to RQ1 is positive.

We further analyze the statistical difference between the measurements obtained on the original images and the beautified ones through paired t-tests on each experiment. The results are shown in Table 1. This test confirms that the distributions are statistically different with p-values below $3.776e - 16$ in all cases.

### 5.3 Experiments on B-LFW: RQ2 - Do beauty filters hinder face recognition?

In this section, we describe experiments to address RQ2, i.e. to shed light on the impact of AR beauty filters on face recognition techniques. Previous works [23, 7] focus on the impact of simple filters on face recognition, particularly filters that apply occlusions of some parts of the faces. However, to the best of our knowledge, there is no previous work analyzing the impact of this type of beauty filters on face recognition. Hence, the analysis of the B-LFW dataset may lead to new insights on

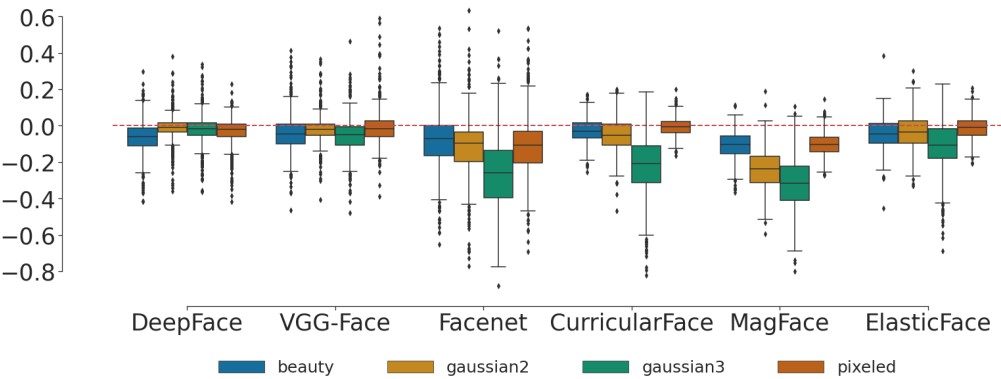

Figure 4: Boxplots of the differences in the distance metric obtained for filtered image pairs versus the metric obtained for the corresponding original pairs of images. A negative value indicates that an image pair was more similar (lower distance metric) when filtered as compared to the original pair. Specifically, each subplot shows these values for the beautified filtered (blue), blurred with a Gaussian filter of radius 2 (yellow) and 3 (green), and down-sampled or pixelated (red) images. The obtained distances and the distances between the original pairs (with no transformation) are first scaled to range [0, 1], then subtracted, to allow a better visualization of the results.

Table 1: Paired t-test results comparing similarity distributions of the original faces and the beautified faces. Each column corresponds to a different sample of 500 couple of images, processed with a different model.

|  | DeepFace | VGG-Face | Facenet | CurricularFace | MagFace | ElasticFace |
|---|---|---|---|---|---|---|
| t-statistic | -15.09 | -8.428 | -10.32 | -9.775 | -30.63 | -11.94 |
| p-value | 1.200e-42 | 3.776e-16 | 9.561e-23 | 9.110e-21 | 1.070e-116 | 4.400e-29 |

understanding the impact of such filters, particularly when no explicit occlusion is applied. This analysis is of societal relevance given the wide adoption of these filters on today's social media platforms.

We evaluate the performance of three state-of-the-art face recognition models (CurricularFace, ElasticFace and MagFace) on the original LFW dataset, on each single beauty filter applied to LFW and on the B-LFW dataset (in which different beauty filters are applied on different images of the same individual). To perform these experiments, we filter the entire LFW dataset [27] with each of the filters, creating eight different variants of it, one for each beauty filter. The obtained results are shown in Table 2, where the filters are shown in the same order as in Figure 2.

Evaluating the impact of each filter on face recognition opens interesting research lines related to studying which properties of AR filters have a stronger impact on face recognition methods. Note how Filter 7 (*big city life* by *triuta*) is the filter that impacts the recognition accuracy the most when compared to the rest of the filters. This effect is consistent across the three state-of-the-art models, as CurricularFace drops performance by $3.74\%$ ($99.80 \rightarrow 96.06$), MagFace by $3.59\%$ ($99.82 \rightarrow 96.23$) and ElasticFace by $3.62\%$ ($99.80 \rightarrow 96.18$). As shown in Figure 2, this filter applies strong modifications not only to the facial features but also to the contrast, hue and exposition of the images.

As previously mentioned, the B-LFW dataset has the purpose of simulating the social media environment, in which different filters co-exist. In Table 2, we observe that the results on B-LFW do not show a significant decrease in the performance of state-of-the-art face recognition models.

Table 2: Verification accuracy (%) of three state-of-the-art models on LFW, eight filtered variants of LFW and B-LFW. Red, Green: respectively, the greatest and lowest performance drop compared to LFW. w/: with. f0 - f7: Filter 0 - Filter 7.

|  | CurricularFace | MagFace | ElasticFace |
|---|---|---|---|
| LFW | 99.80 | 99.82 | 99.80 |
| LFW w/ f0 | 98.93 | 99.47 | 99.17 |
| w/ f1 | 99.33 | 99.42 | 99.50 |
| w/ f2 | 98.90 | 99.37 | 99.35 |
| w/ f3 | 99.13 | 99.45 | 99.33 |
| w/ f4 | 99.13 | 99.45 | 99.43 |
| w/ f5 | 99.18 | 99.49 | 99.67 |
| w/ f6 | 98.08 | 98.42 | 98.38 |
| w/ f7 | 96.06 | 96.23 | 96.18 |
| B-LFW | 99.38 | 99.63 | 99.57 |

## 6 Discussion

In the experiment on the FAIRBEAUTY dataset, we empirically show that, regardless of the selected sample of images and utilized model, there is a general homogenization of the beautified faces when compared to the original ones. However, the experiment on B-LFW shows that the application of beauty filters does not generally impact the performance of the state-of-the-art face recognition models. This result is intuitively consistent with the role of beauty filters in social media: their goal is to improve the appearance of the user while preserving their identity. As a future research direction, we plan to investigate how the homogenization effect of beauty filters varies depending on the similarity between the original pair of images. It is expected that images that are originally different (based on different attributes, such as different ages, genders and/or races) would be homogenized more than images that are originally similar. Investigating this characteristic could highlight intrinsic and subtle biases in the beautification canons of these filters. Note that, in this paper, face recognition techniques are utilized as a research tool to improve our understanding of the impact and behavior of beauty filters, rather than the opposite. We do not conceive our research on beauty filters as a way to improve the quality of current face recognition techniques; in case our readers wish to develop this line of research, we emphasize that they should deeply consider the expected benefits and potential negative consequences of their research. Another direction of future research entails studying the behavior of the beauty filters depending on the facial expressions of the individuals and performing a deeper investigation of the impact of beauty filters on face verification algorithms including different performance metrics beyond accuracy.

The framework proposed in this paper, OPENFILTER, allows researchers from different disciplines to have access to the AR filters available on social media. Despite being flexible and adaptable, we highlight two limitations. First, the framework requires some software skills to precisely follow the given instructions. In addition, due to the resolution limitations of social media, the filters can be applied only to images of up to 512x512 pixels. Unfortunately, this limitation does not allow to fully appreciate the power of some AR filters: beauty filters, for example, apply strong skin smoothing that is less visible on low-resolution images.

We emphasize that any researcher utilizing our datasets should consider their ecological validity before drawing conclusions on the impact of beauty filters on society. As explained in section 2, we have made a significant effort in simulating the real social media environment (further details in the Appendix). However, the users of these platforms tend to use specific communication paradigms –for example, in their poses [54] and facial expressions [42] – which would be represented in a dataset created by scraping the images from social media. However, such practice should be avoided due to privacy and ethical concerns. As a consequence, FairBeauty and B-LFW contain faces that may be demographically more diverse (e.g. in terms of gender) than the faces of the typical users of beauty filters on social media.

# 7 Conclusion

In this paper, we share a framework (OPENFILTER) to automatically apply AR filters on benchmark face datasets. We have also applied popular beautification filters to two publicly available face datasets and have drawn key insights into the characteristics of AR filter-based beautification. We believe that the dynamics related to the effects of beautification filters deserve more interest from different disciplines. We hope that the two datasets shared in this paper (FAIRBEAUTY and B-LFW) and our framework (OPENFILTER) will inspire and support novel research in this field, which is, otherwise, hardly accessible.

## Acknowledgments and Disclosure of Funding

PR and NO are supported by a nominal grant received at the ELLIS Unit Alicante Foundation from the Regional Government of Valencia in Spain (Convenio Singular signed with Generalitat Valenciana, Conselleria d'Innovació, Universitats, Ciència i Societat Digital, Dirección General para el Avance de la Sociedad Digital). PR is also supported by a grant by the Banc Sabadell Foundation. BP is supported by the iToBos project, funded by the EU Horizon 2020 research and innovation program, under grant 965221. FG is supported by the French government, through the 3IA Côte d'Azur Investments in the Future project managed by the National Research Agency (ANR)(ANR-19-P3IA-0002).

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
