# OpenReview forum: "OpenFilter: A Framework to Democratize Research Access to Social Media AR Filters"
_NeurIPS.cc/2022/Track/Datasets_and_Benchmarks — NeurIPS 2022 Datasets and Benchmarks _

### Official Review · Reviewer_PoMm · 2022-07-07
**The new and promising framework for analysis and application of AR filters**

**Rating:** 9
**Confidence:** 4

**Strengths:**

The proposed framework is a novel research instrument for automatic beautification of large datasets. It is outlined that previous user studies consider only small amounts of data and cannot scale effectively. And the existing datasets contain approximations of few AR filters instead of arbitrary user-generated filters.

The authors explain the framework pipeline in detail and provide appropriate evaluation methods for the generated datasets. The framework is described thoroughly and can be used for further research of AR filters applications.

**Weaknesses:**

[resolved] The filters used in the datasets generation and evaluation look very similar, which leads to similarities in quantitative results for the most of them. Hence, the diversity in filters could be improved.

[resolved] The study of the homogenization of faces lacks quantitative results. For more clarity, statistical tests could be reported along with the boxplots.

**Additional Feedback:**

.

**Clarity:**

Lines 154-156: The choice of the beauty filters is based on their popularity, which we assessed through articles in women’s magazines and relevant trends on Instagram.
[resolved] The paper lacks explanation how the sources are credible and not biased, which puts the choice of the analyzed filters in question.

**Correctness:**

The datasets are constructed and evaluated reasonably, the user-generated filters are evaluated against simpler transformations like blurring and rescaling.

[resolved] The choice of filters is questionable: most of them look similar and there is no evidence why more diverse filters could not be used. Therefore, it is not clear if the proposed datasets represent real-world filters well enough.

**Documentation:**

The authors provide a well-documented code repository with both the datasets and the framework available. The data and the code are shared with a non-commercial license to prevent negative unintended effects.

**Ethics:**

The proposed procedure uses the OpenFilter framework to transform two publicly available datasets with existing AR filters, which raises no ethical concerns.

**Relation To Prior Work:**

The authors observe existing datasets and studies, and describe drawbacks of previous approaches which the proposed method aims to overcome.

**Summary And Contributions:**

The paper presents the OpenFilter framework for the application of public AR filters to any set of images, which allows to analyze how AR filters affect models performance and / or human perception. The OpenFilter framework is applied to the publicly available Fairface and Labeled Faces in the Wild datasets, resulting in their beautified versions FairBeauty and B-LFW, which are also published and can be used for further research.

The authors provide a well-documented code repository with both the datasets and the framework available.

---

> ### Author Response · Authors · 2022-08-09
> **Answer to Reviewer PoMm (paper revision 9th of August)**
>
> Thank you very much for your positive and constructive review. Below, we provide an answer to your questions/comments.
>
> **Choice of beauty filters**
> We appreciate your concern regarding the choice of the beauty filters and the possibility that such a choice may have biased our study. Reflecting on your comment, we have added a note in the limitations section of the revised version of the manuscript. However, despite the difficulties in identifying an unbiased choice of filters, we would like to stress that the process was thoughtful and careful, and we did our best to mitigate a potential bias. We have added additional information about the filters and their choice in the Appendix of the revised version of the paper and below in this comment.
>
> Again, we thank you for pointing this out and spurring us to share more information about the filters and their creators.
>
> Hundreds of “beautification” filters created by Instagram users are available on social media platforms. Unfortunately, there is no structured repository of all the available filters and there is no visibility regarding their popularity. Thus, to select a representative sample of filters, we had to rely on information provided by external sources --such as magazine articles featuring the filters--  or on filters created by influential, digital filter creators on Instagram with thousands of followers. Our goal was to capture a representative sample of current beautification filters, trying to mitigate the unavoidable sampling bias related to this choice.
>
> Below, we provide a summary of each filter and Instagram user who created it (the information was last updated on the 2nd of August 2022).
>
> - Filter 0 is called *pretty* and was created by *heru sugiarta*, a digital creator of filters with ~300,000 followers on Instagram
> https://www.instagram.com/herusugiarta/
> - Filter 1 is called *hari beauty* and was created by *hariani*, a digital creator with 10.2 million followers on Instagram
> https://www.instagram.com/hariany/
> - Filter 2 is called *Just Baby* by *blondinochkavika*, a creator of beauty filters with 179,000 followers on Instagram
> https://www.instagram.com/blondinochkavika/
> - Filter 3 is called *Shiny Foxy*, Filter 4 is called *Caramel Macchiato*, Filter 5 is called *Cute Baby Face*, all created by *sasha_soul_art*, an Instagram filter designer of extremely popular filters on Instagram, with 1 million followers
> https://coveteur.com/sasha-soul-interview
> - Filter 6 is called *Baby cute face*, an extremely popular beauty filter created by *anya_ilicheva* with 13,500 followers on Instagram
> - Filter 7 is called *Big city life* by *triutra*, a digital filter creator on Instagram with 138,000 followers
>
> All the users describe themselves as digital creators or digital filter creators and have created several Instagram filters, including the very popular beauty filters used in our study.
>
> **Statistical tests on homogenization study**
>
> Thank you for your comment regarding the lack of statistical tests in the results of Experiment 1. Based on the reviews, we have performed a paired t-test between the distribution of distances in the original faces and in the corresponding beautified case. We have reported the results in Table 1, where it can be observed that there are statistically significant differences between the distributions.

---

> > ### Author Response · Authors · 2022-08-29
> > **Answer to Reviewer PoMm**
> >
> > Thank you very much for taking our changes and responses into consideration. It has been a pleasure interacting with you.

---

### Official Review · Reviewer_gaKC · 2022-07-14
**This paper proposes a novel AR filter-based image transformation framework and answers are required in some respects.**

**Rating:** 7
**Confidence:** 4

**Strengths:**

### Significance
- The proposed framework is valuable in terms of considering all aspects of cultural ecosystems, face aesthetic, and face identity.
- The experimental results that beautified images show lower average distance than original images are encouraging.


### Applicability
- Since the various human-related vision database is built through social media, it seems that the Openfilter framework can be grafted from various perspectives.
- The proposed FairBeauty and B-LFW are easy to access.
- The continued mention of a public license increases the reliability of this dataset.

**Weaknesses:**

- Although the appendix describes the operation of Openfilter framework in detail, it is true that it is difficult to understand the details of the pipeline in Fig. 1 in main body.

- One of the key aspects in social media is the emotional state of the subject. Therefore, the relationship between facial expression recognition and the beautified dataset should be dealt with (at least a brief consideration is necessary).

**Additional Feedback:**

In order to increase the reliability of Table 1, it would be good to use an evaluation metric such as Rank-K in addition to verification accuracy.

**Clarity:**

This paper explains the research background and existing trends really well. It also explains how the Openfilter framework can be used.

**Correctness:**

The explanation of the dataset acquisition process and software script seems reasonable.

**Documentation:**

The operation flow of Openfilter framework can be understood through section 3 of the main body and appendix, but the details in the main body are lacking.
- For example, how Android emulator works, and interworking with virtual webcam, etc.

**Relation To Prior Work:**

A description of the prior work on the beautified dataset seems sufficient. In addition, even the most recent face recognition techniques were used for benchmarking.

**Summary And Contributions:**

### Summary
- This paper proposes an Openfilter framework that visually transforms human faces on social media platforms.
- This paper creates the beautified face dataset FairBeauty and B-LFW.
- Diversity analysis was performed on whether the beauty filter homogenizes the face images. And, correlation between beauty filter and face identity were analyzed.


### Contributions
- AR filters on social media are a novel (research) field, and it is true that they are receiving attention recently in face-related vision tasks.
- Unlike previous filters that are simply superimposed on 2D face images, the proposed framework can create a more realistic and aesthetic face image through the interlocking of AR filter and social media application.
- It deals well with the possibility of extension to various face-related vision tasks.

---

> ### Author Response · Authors · 2022-08-20
> **Answer to Reviewer gaKC (paper revision 20th of August)**
>
> Thank you very much for your positive and constructive review. Below, we provide an answer to your insightful questions/comments.
>
> **Clarity of the pipeline**
>
> Thank you very much for your feedback regarding the pipeline of OpenFilter. Our motivation with the creation of OpenFilter is to enable the research community to perform quantitative research on smartphone-based augmented reality facial filters. Hence, your feedback is very useful. We agree that the original manuscript lacked a very detailed description of the system’s pipeline due to space constraints. Based on your recommendation, we have added proper references in the main text. In addition, we have updated the Github repository and included a more detailed explanation of the pipeline. We have also published a video tutorial that explains how to install and use OpenFilter tool. The video is linked on the Github repository, and can be found here: https://drive.google.com/drive/u/1/folders/16-5wecynAAO5snkwR-2zYpdEpNydIyTx
>
> **Relationship to facial expression recognition**
>
> Thank you for suggesting the study of the relationship between beauty filters and facial expressions. To perform such a systematic study, we would need to apply the beautification filters to benchmark datasets used in facial expression recognition research, such as the AffectNet (http://mohammadmahoor.com/affectnet/) or the FER2013 (https://www.kaggle.com/c/challenges-in-representation-learning-facial-expression-recognition-challenge/data) datasets. Based on your recommendation, we have updated our paper in two ways: we have added this as a potential direction of future work and we included a reference to emotions in Section 6.
>
> **Additional feedback**
>
> Thank you for your suggestion. In our work, we have followed the standard evaluation protocol used with the LFW dataset which consists of reporting the verification accuracy without including any Rank-K metric.
>
> Note that the publications that describe the three face verification systems that we used in our research only report verification accuracies: Tables 2 and 3 in reference [28] which presents the CurricularFace system; Table 1 in reference [36], corresponding to the MagFace paper; and Table III in reference [8] where the authors report verification accuracies on the LFW dataset using the ElasticFace system.
>
> Examples of other well-cited publications that have used the LFW dataset for verification purposes and that only report verification accuracies include:
>
> VarGFaceNet: An Efficient Variable Group Convolutional Neural Network for Lightweight Face Recognition, Mengjia Yan, Mengao Zhao, Zining Xu, Qian Zhang, Guoli Wang, Zhizhong Su, 2019
>
>
> ArcFace: Additive Angular Margin Loss for Deep Face Recognition, Jiankang Deng, Jia Guo, Niannan Xue, Stefanos Zafeiriou, CVPR 2019
>
> VarGNet: Variable Group Convolutional Neural Network for Efficient Embedded Computing, Qian Zhang, Jianjun Li, Meng Yao, Liangchen Song, Helong Zhou, Zhichao Li, Wenming Meng, Xuezhi Zhang, Guoli Wang, 2019
>
> However, a rank-1 identification accuracy metric has been included in publications that have performed their experiments on other benchmark datasets, such as MegaFace and MegaFace(R). Thus, we are grateful for your comment. We believe that a deeper investigation of the impact of beauty filters on face verification algorithms should include different performance metrics to shed light on their reliability. We have added such an evaluation as a line of future work in Section 6 of the revised version of the manuscript.

---

> > ### Comment · Reviewer_gaKC · 2022-08-21
> > **Answer to authors**
> >
> > Thanks for the detailed comments on the weaknesses.
> >
> > 1. In terms of pipeline clarity, the authors' responses and additional tutorial videos were informative.
> >
> > 2. The representative benchmark datasets of facial expression recognition have been well mentioned, and the related responses on Sec. 6 on main body are also agreed.
> > ---
> > Overall, thanks to the improved quality of the paper and helpful discussion, I'll stick with my position.

---

> > > ### Author Response · Authors · 2022-08-29
> > > **Answer to Reviewer gaKC**
> > >
> > > Thank you very much for taking our changes and responses into consideration. It has been a pleasure interacting with you.

---

### Official Review · Reviewer_E3V5 · 2022-07-17
**A method to apply AR filters to images of human faces, and two datasets produced by this method**

**Rating:** 7
**Confidence:** 3
**Clarity:** Yes, the paper is clear.

**Strengths:**

The strengths of the paper (both the framework and the dataset) are summarized as follows:

1. The paper creates a streamlined process to swiftly apply AR filters to images, thus creating an easy way to obtain a dataset with AR filters for research studies that have so far been bottlenecked by the data collection process. The filter is well documented in the supplemental material and in the Github, and the authors provide good instructions for installation, set-up, and even error-handling.

2. The paper demonstrates the usability of the framework by applying it to two renowned datasets, thus avoiding any new ethical issues since the datasets have already been published. The chosen datasets are also unbiased in terms of facial traits and thus any study involving the dataset is truly able to focus on the impact of the AR filters.  The dataset is well documented in the supplemental material with clear instructions on the organization and construction. The authors also provide regular expressions to facilitate easy access and usage of their dataset.

3. The experiments and the results in the paper seem reasonable and reproducible. The paper dedicates a subsection to explain the mathematical foundation behind their experiments. Specifically for the FairBeauty homogeneity experiment, the authors also compare results to gaussian blurring effects and down-sampling, which is a helpful parallel and a reference to interpret the results. Thus, the results seem robust and irrefutable.

4. The problem addressed by the datasets and the framework is relevant to modern society due to the current popularity of AR filters, and should allow for better societal studies on the impacts of social media. The authors themselves are able to identify various areas of research where their dataset can be applicable, demonstrating their thorough knowledge and thought.

**Weaknesses:**

The weaknesses of the paper (both the framework and the dataset) are summarized as follows:

1. The filter, while well documented, does not seem easy to use. There are many steps to be followed in terms of set-up and installation, and the authors themselves acknowledge the need for software skills. The authors intend for their framework to be used in various forms of research that may or may not include software-experienced individuals. Strictly following the long documentation may prove to be intimidating or daunting. The authors could consider posting a demo video of the framework to assist all interested users.

2. For the B-LFW experiment, the authors also provide results on the filtering the entire dataset. The final evaluation and discussion seems to focus on singular filters, which is not the dataset that they are presenting. The results from this experiment thus seem slightly confusing and there is not quite enough information presented on the actual modified dataset, B-LFW. A different form of experiment on the B-LFW dataset, such as perhaps evaluating the average per-person facial recognition performance sectioned by filter, may have been more informative.

3. Although the authors chose unbiased datasets, since all the filters are not applied on all images, there is room for imbalance and bias to be introduced on the portions of the dataset with the filters. The authors do not provide evidence that they maintain the unbiased nature of the original dataset.

**Additional Feedback:**

1. Has the framework been tested on any other device? Curious what comparable speeds are, especially if a GPU is not readily available.
2. What is meant by "heterosexy" in Lines 128-129?
3. How exactly does the selfie culture encourage a "diverse and inclusive" view on beauty? (Line 141)

**Correctness:**

Yes, the claim made by the paper is correct. The dataset construction is thorough and well-detailed, and is sound.

**Documentation:**

The framework and the dataset are both well documented in the supplemental materials, and in the Github repository. Code for evaluation is also provided in the Github.

**Ethics:**

No, the authors use datasets that have already been ethically reviewed and simply apply filters on those datasets.

**Relation To Prior Work:**

Yes, the authors make it clear that the framework is a novelty and there is no existing AR filter dataset.

**Summary And Contributions:**

This paper presents a custom framework called OpenFilter to apply AR filters from social media onto images of human faces. The output of the framework is demonstrated on two well-known and publicly available datasets, FairFace and LFW. The modified versions of these datasets are titled "FairBeauty" and "B-LFW". The authors specifically focus on eight filters that "beautify" faces by some sort of lighting, contouring, or makeup effect, but their framework is able to directly take filters from social media and thus should be able to apply any type of AR filter.

The paper specifically highlights the popularity of AR filters on social media, but the lack of available data for researchers to study the filters and their societal, psychological, and artistic effects. Most filters are applied by users on personal images and is thus unethical to scrape from social media. The only method to obtain AR-modified data so far has been to conduct surveys and user studies, asking users to submit images with AR filters applied to them. The paper thus presents a way to simply apply AR filters to existing human facial image datasets that are already ethically approved.

The authors also demonstrate the applicability of their dataset by addressing two research questions: whether AR filters make all images more similar, and how facial recognition technology performs on their modified dataset. They address the first question with FairBeauty, and the second with B-LFW. They find that the distance between faces does indeed decrease with the applied filters, but facial recognition technology still performs quite well despite the added filters.

The paper also acknowledges that the usage of OpenFilter does require software skills and that there are some resolution restrictions due to social media constraints.

---

> ### Author Response · Authors · 2022-08-25
> **Answer to Reviewer E3V5 (paper revision 25th of August)**
>
> Thank you very much for your positive and constructive review. Below, we provide an answer to your questions/comments.
>
> **Ease of use**
>
> Thank you very much for considering this factor, as our motivation for the creation of OpenFilter is precisely to enable the research community to perform quantitative research on smartphone-based augmented reality facial filters. Thus, we made a significant effort to make the OpenFilter tool as accessible as possible. It is optimized for Windows machines. As the project entails several pieces of software to work together, it is indeed a bit complex. Based on the reviews and to increase the accessibility of OpenFilter, we published a video tutorial to explain the installation and usage of the tool. The video is linked on the Github repository, and you can find it here: https://drive.google.com/drive/u/1/folders/16-5wecynAAO5snkwR-2zYpdEpNydIyTx We have also extensively documented the code to ease its understanding. We hope that OpenFilter will be helpful to any researcher interested in studying AR filters.
>
> **Discussion of B-LFW dataset experiment**
>
> Thank you for pointing out that the discussion of this experiment could be misleading. We agree on this point and we have updated it, explicitly highlighting the results on the B-LFW dataset.
>
> **Impact of applying different beauty filters to FairFaces**
>
> Thank you for sharing your concerns regarding the biases in the creation of the datasets. As explained in the paper, we have made an unprecedented effort to simulate the environment of beauty filters on social media. However, based on the reviews, we have added some considerations on the choice of the filters (Appendix) and the ecological validity of the datasets (Section 6 in the paper).
>
> **"Heterosexy" meaning**
>
> Thank you for pointing this out. The word “heterosexy” is frequently used in the research field of gender studies (including our references). Thanks to your feedback, we have realized that this term may be confusing and thus we have removed it from our manuscript, replacing it with a periphrase.
>
> **Relationship between selfie culture and a diverse and inclusive view on beauty**
>
> Given that any social media user --irrespective of their origin, ethnicity, gender or age-- is able to post their selfies on social media, in theory, social media users would be exposed to a very diverse set of expressions of beauty. For example, in this article: https://www.nationalgeographic.com/magazine/article/beauty-today-celebrates-all-social-media-plays-a-role-feature, we read: _“The millennial generation, those born between 1981 and 1996, is not inclined to assimilate into the dominant culture but to stand proudly apart from it. The new definition of beauty is being written by a selfie generation: people who are the cover stars of their own narrative.”_
> As much as we believe that this statement is arguable and the phenomenon is more complex than this, in the paper we also report this opinion.

---

### Official Review · Reviewer_4dbK · 2022-07-21
**A good paper with a lot of jargon**

**Rating:** 9
**Confidence:** 3

**Strengths:**

Overall this is a well written paper. The experiments in distance and facial recognition are interesting and the authors provide an interesting analysis of some of the ethical considerations, and apply an appropriate open source license to address some of that.

- The data set created by the authors is a valuable addition to the existing data sets for facial analysis
- The analysis done by the others is interesting and valuable in its own right and helps to provide some insight about the filters.


**Weaknesses:**

I don't see any significant weaknesses, outside of the correctness question I raise below.

**Additional Feedback:**

No

**Clarity:**

There are the following minor writing issues:

- There is unnecessary jargon in this paper. An example is the introduction of the word "automatizes" "heterosexy" and so on.
- There are a lot of unnecessary italics in this paper.

**Correctness:**

It seems to be constructed in a sound way. However, the details on which filters were selected and how were a bit sparse. It would be nice to see more information about how the filters were sampled and what the rationale for that sample was.

**Documentation:**

None of the issues around ethical and responsible use are documented on the github site. This should be updated so people who do not read the paper still read about them.

**Ethics:**

I don't se any

**Relation To Prior Work:**

This is not an area of expertise for me and I hope other reviewers will address this.

**Summary And Contributions:**

This paper presents a new data set of filtered selfie images (of faces) constructed by automatically applying a range of filters to two pre-existing face data sets.

---

> ### Author Response · Authors · 2022-08-20
> **Answer to Reviewer 4dbK (paper revision 20th of August)**
>
> Thank you very much for your positive and constructive review. We are very excited to see that you appreciated our work. Below, we provide an answer to your questions/comments.
>
> **Choice of beauty filters**
>
> Thank you for pointing out that information regarding this was a bit sparse in the paper. We have added a new section in the supplementary material to better explain our choices. We report the text here:
>
> Hundreds of “beautification” filters created by Instagram users are available on social media platforms. Unfortunately, there is no structured repository of all the available filters and there is no visibility regarding their popularity. Thus, to select a representative sample of filters, we had to rely on information provided by external sources --such as magazine articles featuring the filters--  or on filters created by influential, digital filter creators on Instagram with thousands of followers. Our goal was to capture a representative sample of current beautification filters, trying to mitigate the unavoidable sampling bias related to this choice.
>
> Below, we provide a summary of each filter and Instagram user who created it (the information was last updated on the 2nd of August 2022).
>
> - Filter 0 is called *pretty* and was created by *heru sugiarta*, a digital creator of filters with ~300,000 followers on Instagram https://www.instagram.com/herusugiarta/
> - Filter 1 is called *hari beauty* and was created by *hariani*, a digital creator with 10.2 million followers on Instagram
> https://www.instagram.com/hariany/
> - Filter 2 is called *Just Baby* by *blondinochkavika*, a creator of beauty filters with 179,000 followers on Instagram
> https://www.instagram.com/blondinochkavika/
> - Filter 3 is called *Shiny Foxy*, Filter 4 is called *Caramel Macchiato*, Filter 5 is called *Cute Baby Face*, all created by *sasha_soul_art*, an Instagram filter designer of extremely popular filters on Instagram, with 1 million followers
> https://coveteur.com/sasha-soul-interview
> - Filter 6 is called *Baby cute face*, an extremely popular beauty filter created by *anya_ilicheva* with 13,500 followers on Instagram
> - Filter 7 is called *Big city life* by *triutra*, a digital filter creator on Instagram with 138,000 followers
>
> All the users describe themselves as digital creators or digital filter creators and have created several Instagram filters, including the very popular beauty filters used in our study.
>
> **Clarity**
>
> We agree about the excessive use of italics and ambiguous terms. We have significantly reduced their use in the revised version of the manuscript. The word “heterosexy” is quite used in the research field of gender studies (including our references). Thanks to your feedback, we have realized that this term may be confusing and thus we have removed it from our manuscript, replacing it with a periphrase.
>
> **Github ethics**
>
> Thank you for pointing this out. As our aim is to promote ethical and responsible research, we have updated the Github page with an explicit disclaimer on the intended uses and have restricted the commercial use of our software to the maximum allowed by an open source license.

---

### Official Review · Reviewer_Q8Qr · 2022-07-22
**Potential to increase access to hard-to-study aspect of social media; ethically contentious area requiring careful consideration; some fixable methodological flaws**

**Rating:** 7
**Confidence:** 4
**Clarity:** Clearly written, good organization, n…

**Strengths:**

•	well-written

•	authors make a fairly convincing case that their tools could help researchers access, probe, and understand a usually proprietary and inaccessible feature of the contemporary social media landscape – “democratizing” access, as they say.

•	motivating questions, technical set-up of the two illustrative experiments were easy to grasp, though the analysis of experiment 1 is flawed (see weaknesses).

•	authors do a good job of situating their technical contribution within the societal and broader research context: provide some background on selfies and AR filters on social media, mention  potential research topics that could benefit from their tool and datasets, and discuss ethical issues.


**Weaknesses:**

The authors take some good steps to address the ethical stakes of this research. Still, some important considerations are unaddressed – see ethics section of review.

The analysis and presentation of experiment 1 data is flawed – see correctness section of review


**Additional Feedback:**

A few miscellaneous questions and thoughts:


1. Has the apply-each-filter-to-LFW dataset from experiment 2 also been released? This isn’t clear to me.

2. On pg. 4, there are some potentially contentious claims about how beauty filter transformations align with different ethnicities’ associated characteristics – are these fairly well established? Are they the authors’ own impressions? I notice there is no source cited for this.

3. The discussion of beauty standards mainly speaks of femininity and female faces. Do we know anything about the balance of sexes present in the benchmark datasets or the filters used? Any existing research on filter choices and how it might be gendered? Are all the filters ‘feminine’ oriented or are there also masculine filters included among the beauty filters? Potential implications for research?

4. In addition to the technical limitations mentioned in the Discussion, it is worth considering  remaining limitations to ecological validity or what can be studied with the data. For example, we do not know whether the photos to which beauty filters have been applied here actually reflect the kinds of photos (poses, expressions etc.) people tend to use beauty filters on. That is: the dataset doesn’t reflect actual human choices to apply or not to apply beauty filters in real world settings. That seems worth emphasizing, as it means we cannot use these data to make claims like, "people tend to apply filters to photos with characteristics X".

Overall, with fixes to experiment 1’s analysis and some more discussion of the ethics around facial verification, I would recommend for this to be published.


**Correctness:**

I cannot really technically evaluate the Open Filter tool itself, but the set-up and dataset creation processes sound plausible. The experimental designs make sense for investigating the authors’ questions about homogenization and face verification performance. However, the approach to analyzing experiment 1 is at best, suboptimal.

In experiment 1 (Algorithm 1), the authors :

* randomly sample pairs of images from FairFace
* apply four different ‘treatments’ (beautification, blurring 1, blurring 2, pixelation) to the images
* for the originals and each of the treated pairs, calculate the distance between the pairs of faces.

The idea is that if the distance between pairs of beautified faces is lower than for the originals, this is evidence that the beautification filters create more homogeneity in the faces.

This is essentially a blocked experimental design *at the level of the (dis)similarity measurements* -- that is, we have groups of similarity measurements which are all for the same pair of original images. However, in their analysis, the authors ignore this paired design and simply calculate the average distance for each treatment group and look at the differences in group averages. This is an inefficient way to analyze these data that throws away information and results in less power to detect a treatment effect.

Essentially, this is the difference between doing a two-sampled t-test (is the **difference in the means** of two independent samples different from 0) and doing a paired t-test (is the **mean difference** different from 0?). When we have paired data but act as if they are not paired, we ignore the fact that observations from the same pair may tend to be more similar (intraclass correlation). In the case of experiment 1, distance measurements that are all applied to images derived from the same baseline sets of faces will presumably tend to be more similar to each other than distance measurements randomly selected across baseline sets of faces. Looking at the mean difference controls for that varying baseline by asking, “given each baseline set of faces, are $x_b$ and $x'_b$ more similar than $x$ and $x’$?” while looking at the difference in means does not account for the varying baseline and thereby introduces more noise into the comparison. Comparing the group averages gives some indication, but it’s not the best you can do.

Instead of the current analysis, I recommend that the authors look at the pairwise differences in similarity for each of the treatments compared to the original. In the paper’s notation, for each randomly sampled pair ($x$,$x'$ ), one would calculate $∆_b=m_b-m$,  $∆_g'=m_g'- m$, $∆_g''=m_g''-m$, $∆_s=m_s-m$. I recommend the authors then use the boxplot approach of Figure 4 to show the distribution of $∆_b$,$∆_g'$,$∆_g''$, and $∆_s$ (note: there would be no boxplot for the original group anymore since the others are relative to it). If, on average, the $∆_b$'s are less than 0, this indicates that the beautified faces tend to be more similar (feel free to flip it so positive indicates effect). You could also do pair t-tests and report the results – this will allow you to do more than say the result is “positive.” Doing all this could still yield a result that suggests homogenization – it could even make the case more strongly.

A few other things on experiment 1:

* I see no reason to sample different sets of 500 images for each of the models. This only introduces an additional source of variation that hinders our ability to compare the models. Ideally, you could apply all the models to 3000 images but if this is infeasible, I would prefer all models applied to the same 500
* Figure 4 is not accessible to those who are color blind. This could be improved by providing a label or at least stating in the caption that the order of the key at the bottom is also the order of the box plots
* It would be good to report a table with numbers for experiment 1
* Something you might look at or mention as a future extension: it would be interesting to examine whether $∆_b$ varies with the baseline distance of the original untreated images (could plot a scatterplot). I imagine there could be a saturation effect. For example, take faces of two siblings of similar age and same sex and beautify them and the original images might already be so similar that there isn’t much left to increase. Take photos of people of different ages, sexes, and/or ethnicities and there could be much more room to homogenize.


**Documentation:**

Documentation in the supplement seems useful. The authors explain more about how their tool works and describe how the file systems work for each of their beautified datasets in relation to the originals. Although I have not tried installing the tool myself, it was easy for me to at least find instructions on how to do so. I was also easily able to find where to download the datasets.

Licensing information is included.


**Ethics:**

The authors frame the research as an alternative to the more unethical step of scraping images off of social media without people’s consent. Still, this research deals with data in an ethically contentious domain and the datasets released include personally identifiable information. Although the researchers discuss ethical issues, some issues are under-addressed and should be either reviewed or more discussed in the paper. There are at least three types of ethical issues present


**Consent**

* Did the people in the face datasets consent to their faces being present in these datasets? it is unclear from the paper. The datasets used are open source, published datasets, so primary responsibility does not lie with the current authors. Still, it is important to mention under what terms of consent (or lack thereof) the FairFace and Labeled Faces in the Wild datasets were collected so that this information is not buried along the pipeline.

* Did the particular people displayed in the paper consent to their faces being presented in the paper in this way? If not, and if it is no longer possible to contact those people, it seems to me that it would be more ethical to use other example photos for which there is consent. That said, I do not know whether the current ethical consensus on this issue differs from my own intuition – I have seen other face recognition related papers make sure to note consent for highlighted photos.

**Bias**

The authors address this issue well. They recognize that beauty filters have potential to exacerbate biases (e.g., colorism) --  but this is not so much a hazard of their dataset as of the phenomenon that their dataset could help researchers to study. That is: as the authors recognize, their tool and dataset could help researchers discover biases.  The authors also chose to use FairFace as the basis for one of their datasets exactly because it is meant to be a dataset of diverse faces – so while there is no guarantee that representation issues are perfectly addressed, they were taken into consideration. The coverage and biases of the LFW dataset are not mentioned.

**Uses**

The authors warn that they “discourage controversial and unethical of our framework and datasets” and say they publish under a non-commercial license to try to protect the public from negative uses of the dataset. This does not of course, guarantee misuse won’t happen. The authors at least seem cognizant of the issues, and it does seem like the dataset and tool could have benefits. Still, I cannot fully evaluate on my own whether the benefits outweigh the risks. I probably do support publishing this research because of its ‘democratizing’ role in allowing more people to try to understand characteristics of social media behaviors at scale.

The authors highlight “developing beauty removal applications” as an example of  potentially unethical use, but I think there is an additional ethically fraught use that seems under-discussed: facial verification. Indeed, experiment 2 applies face verification but the authors do not mention that this use case can have implications for privacy and anonymity (if certain filters DID lower verification accuracy, this kind of tool might be used to train an algorithm to overcome that) and is used in high stakes contexts (e.g., policing, banking, security) in which some people argue against the further development of this technology at all. The potential for this tool to be used to make facial verification more robust seems to me a graver ethical risk of releasing this dataset than the development of beauty removal applications. There is still an argument to be made that this dataset also provides opportunity for people to understand how filters might obfuscate – or not – their identities in contexts where they do/don’t want their faces to be verifiable... and face verification is not always harmful...but realistically, it is likely powerful actors will benefit more from this kind of tool than everyday people. I would like to see the authors grapple with this more rather than treat facial verification as a straightforward use of the dataset.


**Relation To Prior Work:**

Authors emphasize that while other research has worked with simpler filters that cover some part of the face, their tool and datasets better reflect the real-world social media ecosystem and therefore could support more ecological validity

**Summary And Contributions:**

The paper considers Augmented Reality filters for faces, which are used on many social media sites. The authors argue these filters are of interest to scientific researchers from psychological, social, cultural, and artistic perspectives but that researchers are currently limited in their ability to study these filters at scale because they are often proprietary and cannot ethically be scraped from the platforms. To provide researchers with access to real-world AR filters from social media, the authors have developed a tool that can apply AR filters to an existing dataset of faces. The authors also contribute two new datasets that result from applying a collection of 8 popular beauty filters to two existing open access face datasets: FairBeauty (applied to FairFace) and B-LFW, applied to Labeled Faces in the Wild (LFW). These are meant to illustrate potential uses of their tool but also provide resources for studying the role of beauty filters specifically. The authors present two example experiments: One which compares FairBeauty to FairFace to look at whether beauty filters make faces more homogeneous (lowers distance between randomly selected pairs) and one which uses LFW and B-LFW to study effect of beauty filters on accuracy of facial verification algorithms (they find little decline in performance).

---

> ### Author Response · Authors · 2022-08-09
> **Answer to Reviewer Q8Qr (paper revision 9th of August) pt.1**
>
> Thank you very much for your positive and constructive review. Below, we provide an answer to your questions/comments.
>
> **Recommended improvements to Experiment 1**
>
> Thank you for your precious suggestions on the plot of Experiment 1. We agree that the previous version of the paper presented a Figure with unnecessary noise. We proceeded in updating it accordingly. Regarding the color palette, we now utilize the *colorblind* palette provided by the library *seaborn*. In addition, we have performed paired t-tests between the distributions of the measurements on the original case and on the beautified case. We have reported the results in a Table, confirming the statistical difference between the distributions.
>
> We understand your concern regarding the choice of different samples for each model. We have slightly changed the comment to the results, clarifying that we do not intend to make a comparison between the six models, and the only intent of Experiment 1 is to show that regardless of the chosen model and the chosen sample, the beautified images result to be more similar than the original ones. For this reason, we think it is interesting to have different samples on different models, so that it would not be possible to argue that the behavior is due to the chosen sample of images.
>
> Thank you for your suggestion regarding a study of the potentially different homogenizing impact of the filters depending on the similarity of original pairs of images. As you have noted, the homogenization effect could be much larger in cases of pairs of faces corresponding to different ages, genders and ethnicities. We believe that this could be an interesting starting point for future work analyzing the impact of these filters on different facial attributes. We have updated Section 6 (Discussion) of the revised manuscript accordingly.
>
> **Ethical implications for face verification**
>
> Thank you for emphasizing the ethical implications of our work as our focus is precisely to enable research that would consider the wider societal  implications of AI-enabled technology.
>
> We agree that face verification is a controversial application of computer vision techniques that needs to be discussed further before being developed. Indeed, we did not use our datasets to re-train any of the state-of-the-art methods. In the case of our research, rather than utilizing the dataset as a tool to improve face recognition, we used face recognition as a tool to improve our understanding of the characteristics of beauty filters. However, we understand that other scientists and practitioners in the field could have a different perspective on this. Based on your review, we have updated both Section 4.1 and 6 in the paper, stating our take on this technology more clearly. In addition, we have also strengthened the licensing model of our code and we have inserted a disclaimer in the code repository, explicitly requesting that anyone using our code or assets should consider the ethical implications and minimize potential negative consequences.
>
> **Consent**
>
> Regarding your concerns on the consent issues for these dataset, we point out that:
> - The *FairFace* dataset is publicly available with a license CC BY 4.0. This license enables sharing, copying and redistributing the material in any medium and format and adapt, remix, transform and build upon the material for any purpose, even commercially. Hence, we had permission to create the *FairBeauty* dataset as derivative work from the FairFace dataset. According to paperwithcode.com, over 60 publications have used the FairFace dataset since 2019. The YFCC-100M dataset that FairFace is based upon is a large-scale publicly and freely usable multimedia collection, containing metadata of around 99.2 million photos from Flickr which were shared under a Creative Common license. This dataset was produced at Yahoo Labs by Bart Thomee and David Ayman Shamma, with the assistance of Li-Jia Li and Nikhil Rasiwasia, as well as the Yahoo Webscope and Legal teams.
> - *LFW* (Labeled Faces in the Wild) is a publicly available benchmark dataset, mainly used for face verification purposes. This dataset contains more than 13,000 facial images of 1,680 different individuals. According to paperswithcode.com, this dataset has been used by ~400 publications since 2018. The original images used in LFW were extracted from: Tamara L. Berg, Alexander C. Berg, Michael Maire, Ryan White, Yee Whye Teh, Erik Learned-Miller, and David A. Forsyth. Names and faces in the news. CVPR, 2004. This CVPR publication presents a dataset of half a million news pictures and captions from Yahoo News collected over a period of roughly two years. Thus, all the faces included in LFW correspond to faces of public figures (e.g politicians, artists, sports personalities, etc...) that appeared in the news over such a time period.
>
> (continues in pt. 2)

---

> > ### Author Response · Authors · 2022-08-09
> > **Answer to Reviewer Q8Qr (paper revision 9th of August) pt.2**
> >
> > Regarding the consent in using the pictures in our paper, thank you for pointing it out. Given the public nature of the datasets that we have used, the common practice in related publications that have also analyzed these datasets is not to explicitly report obtaining explicit consent for inserting the images in the papers. As examples, we report:
> >
> > - https://arxiv.org/pdf/2103.09118v3.pdf
> > - https://arxiv.org/pdf/2007.06141v1.pdf
> > - https://arxiv.org/pdf/2101.04061v2.pdf
> > - https://arxiv.org/pdf/2104.09874.pdf
> >
> > **B-LFW characteristics**
> >
> > Thank you for pointing out the confusion related to the available versions of this dataset. In B-LFW the eight different filters are applied in equal proportions to the dataset, making sure that different photos from each individual are beautified with different filters. However, for research purposes we also share the eight different versions in which all the images are beautified with the same filter. We have clarified this in the Appendix.
> >
> > **Feminine focus of filters // ecological validity // characteristics of beauty filters**
> >
> > Thank you for pointing out these relevant questions regarding the gendered aspect of the filters, the ecological validity of the datasets and the canon of beauty behind the beauty filters. We believe these three comments are part of the same issue. Note that there is no structured repository with all the available filters on Instagram and thus it is extremely difficult to collect statistics about their use.
> >
> > To shed light on this topic, we performed search queries with relevant hashtags, such as #beautyfilter and related keywords on Instagram (both among posts and filters). In a qualitative and approximated manner, our searches revealed that the majority of users posting beautified content are women.
> >
> > Gender biases related to beauty are, to some extent, intrinsic in our society (and not only in the beauty filters available on social media) but we could not measure such a bias directly on Instagram or other social media platforms by scraping online images for ethical reasons.
> >
> > Nevertheless, we followed a systematic and rigorous approach to select representative beauty filters, based on the popularity of the filters on the platform, assessed through external sources. Based on the reviews, we have updated both the discussion paragraph (regarding the ecological validity) and the Appendix of the revised version of the paper (regarding the choice of filters).
> >
> > Concerning the characteristics of the beauty canons behind the filters, we did not insert a reference because that sentence was a summary of our literature review of different sources and also a result of our extensive study of the filters. We understand that the sentence could sound contentious because of the estimated cultural references of the facial traits that get modified by the filters. Thus, based on your feedback, we have rephrased the sentence and we have included several references from which interested readers can learn about the cultural issues behind beauty filters.

---

> > > ### Comment · Reviewer_Q8Qr · 2022-08-20
> > > **Mostly satisfied with changes made - the paper has improved**
> > >
> > > Thank you for responding thoughtfully and extensively to my review. I am glad to see that you:
> > >
> > > * removed the references to ethnicity and just described the traits beauty filters tend to emphasize - it reads much better
> > >
> > > * added in some more detail on where the faces in the wild dataset comes from
> > >
> > > * added notes on ecological validity to the discussion section
> > >
> > > * referred more to the ethical issues I mentioned -- these still feel rushed/brief in places and as another reviewer states, ideally, this paper could be accompanied by more interdisciplinary discussion and critical reflection to model what fully-developed research using this kind of tool should look like. There remains the uncomfortable reality that the data/tool *could* potentially be used for less ethical purposes and that some readers might skim over the ethical cautions. That said, this is not a problem that I hold this paper responsible for on its own and I think the authors have made some improvement to their discussion and licensing.
> > > ----
> > >
> > > I am also glad to see the change to Figure 4 and the addition of the paired t-test. **Note, however, that the caption to Figure 4 has not been properly updated - it still says the boxplots are of distances rather than differences in distances. Please fix this.** I would have it read something like the following (feel free to use in whole or in part):
> > >
> > > “Boxplots of the differences in the distance metric obtained for filtered image pairs versus the metric obtained for the corresponding original pairs of images. A negative value indicates that an image pair was more similar (lower distance metric) when filtered as compared to the original pair.  Specifically, each subplot shows these values for the beautified filtered (blue), blurred with a Gaussian filter of radius 2 (yellow) and 3 (green), and down-sampled or pixelated (red) images" <and then the part about scaling>
> > >
> > > On the question of using different n = 500 samples of images, I suppose I am ok with leaving it as it is but for the future, I would note that the best way to address the issue of results being too dependent on the particular sample drawn or on outlier cases is not to use 500 entirely different samples (introducing a form of uncontrolled variation into the comparison) but to increase the sample size. By putting the different models on the same plot you are inevitably inviting comparison and it would be nice to know that the fluctuations between models are not attributable to chance differences in the images selected.
> > >
> > > ---
> > >
> > > One typo note:
> > >
> > > * Line 144: “as every technology” should probably be “as with every technology”
> > >
> > > ---
> > >  I will increment my rating from a 5 to a 7 but not further because I am still a little uncomfortable with the ethical issues (but not enough to recommend against publication entirely)

---

> > > > ### Author Response · Authors · 2022-08-29
> > > > **Answer to Reviewer Q8Qr**
> > > >
> > > > Dear Reviewer,
> > > >
> > > > Thank you very much for your constructive feedback and for taking into consideration the changes that we implemented in the revised version of the manuscript.  As per your recommendation, we have updated the caption of Figure 4 and have corrected the typo that you detected. Thank you very much!
> > > >
> > > > We are really grateful for having had an interactive and constructive review experience with you. Thanks for sharing your wisdom with us, which has undoubtedly helped us improve the manuscript.

---

### Official Review · Reviewer_jwSG · 2022-07-25
**Social impact of dataset not sufficiently addressed**

**Rating:** 3
**Confidence:** 4
**Clarity:** The work is clearly described.

**Strengths:**

1. Significance: the authors identify a social phenomenon which has lacked quantitive research due to a lack of data. By producing a large-scale dataset, further research into social media filters and their social impact could be enabled.

**Weaknesses:**

1. Sensitivity to ethical and social implications: described in more detail in the Ethics section below.
2. Accessibility: even though the stated intent of the work is to enable interdisciplinary research into social media filters, reproducing the datasets requires non-trivial computer science skills.

**Additional Feedback:**

If the intent of this work is to enable quantitative, interdisciplinary research regarding the properties of social media filters, and how they impact social subgroups, then I strongly suggest to include that as part of the work given the sensitive nature of the dataset.

**Correctness:**

The construction of the dataset is sound. Conclusions from the 2 experiments seem correct.

**Documentation:**

The data collection is clearly described in the paper. The intended use section describes the 2 research questions as explored in the paper. Licensing for the software and dataset are clear. Details on hosting and maintenance seem to be missing.

**Ethics:**

This dataset purposely contains human biases as encoded in the social media filters that were applied. The authors recognize some of the known issues: "the extensive use and exposure to beautified selfies seems to be a homogenizing force of beauty standards, contributing to a significant increase in teenage plastic surgery and mental health issues", and "These filters have raised concerns regarding existing biases in the automatic beautification practices and have been widely criticized for perpetuating racism and, in particular, colorism".

The authors discourage non-ethical use by publishing the dataset under a non-commercial license. However, the code to reproduce is licensed under GPLv2 which allows commercial use, going against the intention of discouraging non-ethical use.

Regardless of the dataset and software licenses, the work lacks a proper assessment of all social biases represented in this dataset. The authors consider this as part of future work: "In this paper, we do not take a stand on the virtues and dangers of beauty filters for society: as every technology, its use creates a spectrum of possibilities whose value should be properly investigated and understood through interdisciplinary research". Releasing this dataset without such interdisciplinary research, however, introduces the risk of unethical use of a dataset that is biased by construction.

**Relation To Prior Work:**

Yes, the authors discuss the need for quantitative research into social media filters and how the construction of this dataset enables that.

**Summary And Contributions:**

The authors publish a tool for applying social media filters to facial images, as well as 2 datasets to which this tool was applied. The datasets FairBeauty and B-LFW were generated by applying the OpenFilter tool to the pictures from the open source datasets FairFace and LFW, respectively.
The paper describes 2 experiments, one for each dataset. The FairBeauty dataset was used to investigate the similarity of faces post-application of the social media filters. The B-LFW dataset was used to compare the performance of face recognition algorithms pre- and post-application of the social media filters.

---

> ### Author Response · Authors · 2022-08-09
> **Answer to Reviewer jwSG (paper revision 9th of August) pt.1**
>
> Thank you very much for your constructive review and for emphasizing the ethical implications of our work as our focus is precisely to enable research that would consider the wider societal  implications of AI-enabled technology. Below, we provide an answer to your questions/comments.
>
> **Accessibility of the OpenFilter Tool**
>
> Thank you very much for considering this factor, as our motivation for the creation of OpenFilter is precisely to enable the research community to perform quantitative research on smartphone-based augmented reality facial filters. Thus, we made a significant effort to make the OpenFilter tool as accessible as possible. It is optimized for Windows machines. As the project entails several pieces of software to work together, it is indeed a bit complex. Based on your feedback and to increase the accessibility of OpenFilter, we published a video tutorial to explain the installation and usage of the tool. The video is linked on the Github repository, and you can find it here: https://drive.google.com/drive/u/1/folders/16-5wecynAAO5snkwR-2zYpdEpNydIyTx
> We have also extensively documented the code to ease its understanding. We hope that OpenFilter will be helpful to any researcher interested in studying AR filters.
>
> **Hosting and maintenance plan**
>
> Based on your review, we have extended the Appendix. We report the revised  text here:
>
> The project is version-trackable on our Github repository (https://github.com/ellisalicante/OpenFilter), where it will be permanently available. The datasets FairBeauty and B-LFW are hosted on Microsoft Azure, from where they can be downloaded. The dataset was created at the ELLIS Unit Alicante, and the authors are committed to maintain the repository and the dataset storage at least until 2025, providing proper maintenance and development. Piera Riccio is in charge of supporting, hosting and maintaining the dataset. She can be contacted at her email address: piera@ellisalicante.org.
>
> For the time being, the authors do not foresee periodic updates of the dataset, but it will be corrected in case any error is detected in the current version. The availability of older versions will be subject to the type of update that is performed. The authors will make sure that any update is clearly communicated and justified to the rest of the community through the official GitHub page and the project page (https://ellisalicante.org/datasets/OpenFilter).
>
> If other researchers are interested in collaborating on this work by extending or augmenting the datasets, they are warmly encouraged to get in touch with the authors. The authors will evaluate each proposal for extension before including it in the dataset. Even in this case, the authors will make sure to properly communicate the updates on both the GitHub and the project's page.
>
> **Ethical concerns: code release license / social biases represented in the dataset**
>
> Thank you for your keen observations regarding the ethical implications of our work. Indeed, enabling the systematic study of ethical and societal implications of AR face filters is the main motivation for our work.
>
> Regarding the license of the code, NeurIPS asks for all code to be released using an open source license. Hence, we have released the code using a common open source license which allows commercial uses, as all open source licenses do. However, based on your recommendation, we have investigated further the options that open source licenses offer to limit commercial use.
>
> According to our research, a dual licensing model seems to be the most suitable option. Thus, we have changed the license of our code accordingly, inserting an explicit exception of usage for commercial purposes. Note that we have also included in the header of our code an explicit request to always consider the ethical implications of the technology and to avoid unethical uses or applications.
>
> In addition, we have changed the license for the two datasets from CC BY-NC to CC BY-NC-SA, forcing any derivative work to be non-commercial as well.
>
> With respect to the social biases embedded in the beauty filters, the main motivation for creating OpenFilter and releasing the two beautified face datasets is precisely to enable and encourage the research community to explore such biases. Without OpenFilter, quantitative research to address these issues is not possible, which explains the scarcity of scientific work on this topic. We hope that through the analysis of FairBeauty and B-LFW, scientists will be able to shed light on the social biases embedded in the beauty filters. Section 4.1 in the paper clearly states that the intended use of OpenFilter and the datasets is precisely to explore such research directions.
>
> (continues in pt.2)

---

> > ### Author Response · Authors · 2022-08-09
> > **Answer to Reviewer jwSG (paper revision 9th of August) pt.2**
> >
> > Moreover, we believe that the public release of OpenFilter and the two beautified datasets will facilitate interdisciplinary research on the topic, as the datasets are already created and could be used in a variety of user studies to shed light on the influence of beauty filters on cultural constructs. Scholars in other fields of research (e.g. social sciences, anthropology, aesthetics, psychology) are already investigating the impact of beauty filters on our society, providing insightful perspectives to such an important debate. However, it is currently very difficult for researchers in computer science-related fields to support such research with quantitative analyses. The main reason is the lack of data and the difficult access to the filters, which are precisely the main contributions of this paper: OpenFilter to enable access to the filters and two beautified face datasets. That’s why we submitted it to the datasets and benchmarks track.
> >
> > Beyond the tool and the datasets, we illustrate the type of research that our datasets enable via two experiments. In Experiment 1, we provide insights on the homogenization caused by these filters, supporting with a quantitative analysis what other researchers have observed in qualitative research settings. To the best of our knowledge, no similar attempt was done before.
> >
> > A thorough analysis of the actual social implications of these filters is a necessary and important research direction that our datasets enable and foster. However, we believe that such a thorough analysis is out of the scope of this specific NeurIPS publication, focused on presenting the OpenFilter tool and the
> > datasets.

---

> > > ### Comment · Reviewer_jwSG · 2022-08-19
> > > **Answer to authors**
> > >
> > > Thank you for addressing the accessibility concerns and clarifying the hosting and maintenance plan.
> > >
> > > Comments:
> > > 1. The walkthrough video is helpful and shows the willingness of the authors to support derivative work.
> > >
> > > 2. The software license has been adjusted, but still allows commercial use on request. If the motivation of the work is "enabling the systematic study of ethical and societal implications of AR face filters", why still allow for potential commercialization?
> > >
> > > 3. I agree with the authors that this dataset could enable quantitative research. I still believe that this should be an interdisciplinary work from the start, considering thorough analysis of social biases in scope for the original work.

---

> > > > ### Author Response · Authors · 2022-08-20
> > > > **Answer to Reviewer jwSG**
> > > >
> > > > Dear reviewer,
> > > >
> > > > Thank you very much for your feedback to our response. We are happy to see that you appreciate our efforts with the walkthrough video.
> > > >
> > > > Regarding the software license, we have opted for the most restrictive mechanism that is allowed by an open source license. If you are aware of an alternative solution to restrict even further the commercial use of the software while releasing it as open source, we would very much appreciate your guidance on this regard. Note that both NeurIPS (https://neurips.cc/Conferences/2022/CallForDatasetsBenchmarks) and our research institute follow an open science research paradigm which includes making our research code available as open source.
> > > >
> > > > Finally, we agree with you regarding the need for interdisciplinary research in the context of AR filters. However, such research is right now very difficult --if not impossible-- to do due to a lack of (1) tools to apply AR filters to pre-existing research image datasets and (2) benchmark datasets that have been transformed with such AR filters. Enabling such research is the main motivation for our work. Beyond the two research questions addressed in the paper, section 4.1 outlines two research directions that our datasets enable and that require interdisciplinary research: first, the study of the influence of beauty filters on social constructs, such as trustworthiness, homophily and intelligence, both computationally and through user studies; second, societal implications of beauty filters, including potential racial biases in the filters.
> > > >
> > > > We see our contribution with this manuscript as the first necessary step to enable such research.

---

### Official Review · Reviewer_oHUJ · 2022-08-22
**Generation and analysis of open sets of 'beautification' filter images**

**Rating:** 8
**Confidence:** 4

**Strengths:**

The paper clearly describes the methodology and results. The strongest contribution is in the analysis of the comparative homogenization of faces and the analysis of how filters affect facial recognition.

The datasets of filtered facial images may be of interest to other researchers.

**Weaknesses:**

The method of scripting use of online platforms to apply filters may not be reproducible. The authors automated application of filters to run them on up to 22,000 images per day but did not report working with the platforms to ensure that this was an acceptable use - if platforms begin to limit the number of applications per day per user, this technique will no longer work.

The authors rightly note that 'beautification' filters are constantly changing and rising and falling in popularity. Their datasets use several of the most popular filters at the time of writing, but this will quickly date the work as new filters become popular over time. The intent is to allow other researchers to use the open-source toolset to create new datasets with new filters, but this relies on some degree of cooperation (or at least non-interference) by the platforms being used to run the filters across large numbers of images.

It is not clear whether the use of scripting to automate filtering thousands of images is allowed under the terms of service of the various platforms.

**Additional Feedback:**

none.

**Clarity:**

In table 1, the p-values are certainly low, but more discussion would help to clarify the significance of this result. There is clearly a statistically sound difference between the values, but it's not clear whether it's a meaningful difference.

**Correctness:**

The claims and analysis seem sound - another reviewer remarked on some methodological concerns about the use of t-tests and these seem to have been addressed.

**Documentation:**

Yes, datasets are documented and clear.

**Ethics:**

The platforms that the authors used to apply the filters to the images in the dataset are very large and handle huge amounts of traffic. Running 22,000 images per day is a drop in the bucket compared to their daily totals. Even so, it is not clear whether scripted use of the service to create an open dataset is acceptable under the various platforms' terms of service. The authors should clarify that it is acceptable as this is a cornerstone of their approach.

**Relation To Prior Work:**

Yes, the authors are clear where they build on and extend prior work.

**Summary And Contributions:**

The paper introduces a method for applying popular 'beautification' filters to an open dataset of facial images. This results in further datasets of modified images (one for each filter). The authors use these datasets to analyze the amount to which the filters homogenize faces and whether the filters affect the accuracy of facial recognition algorithms. The datasets and code are released under GPL and CC licenses.

---

> ### Author Response · Authors · 2022-08-25
> **Answer to Reviewer oHUJ (paper revision 25th of August)**
>
> Thank you very much for your constructive and positive review. Below, we provide an answer to your main question/comment regarding the automatic application of existing filters on thousands of images.
>
> Before developing OpenFilter, we devoted a non-negligible effort to address the issue that you mention. We did not identify any restrictions on Instagram’s terms and conditions, and did not find any limit to the application of filters on existing images (https://help.instagram.com/581066165581870). Moreover, users are allowed to apply a few of the available filters to existing images (https://www.techniquehow.com/add-instagram-filters-to-existing-photos/), which is what we did to generate the beautified versions of the  two face datasets, overcoming technical limitations on most of the available filters.
>
> Regarding the possibility of the platforms limiting the number of applications per day, note that in the current version of OpenFilter, the different images are projected on the camera simulating a live stream video. The filters are applied on every frame, but it would be the same if we just opened the camera and started taking a video of ourselves. We do not predict, for the time being, that the platforms would limit this functionality, but if that would be the case, we will make sure to update the framework (if possible) or to notify other users about it.
>
> Hence, we do not anticipate any issues with OpenFilter as long as the images where the filters are applied are openly available and not subject to copyright or privacy restrictions. We also include the necessary attribution to the creators of each of the exemplary beauty filters that we selected to create the datasets.

---

### Author Response · Authors · 2022-08-29
**General comment from the Authors**

We would like to thank all the reviewers for their constructive and extensive feedback that has certainly helped us improve the manuscript. We are also thankful that the reviewers have interacted with us during the discussion period. Most of them have responded to our comments with insightful contributions.

---

### Meta-Review · Program_Chairs · 2022-09-16

**Recommendation:** Accept
**Confidence:** 4

**Metareview:**

This paper contributes a tool as well as a dataset to study the application of social media AR filters to enable interdisciplinary research  from psychological, social, cultural, and artistic perspectives. The reviews are largely positive (6 positive, 1 negative) and the authors have successfully  addressed some of the weaknesses that the reviewers raised, eg. the reviewers addressed the concern that the tool is non trivial to use without computer science skills, though a tutorial video. The authors have also addressed the ethical concerns raised by the reviewers. I think this paper meets the bar for the track and should be accepted

---

### Decision · Program_Chairs · 2022-09-16

Accept